# Self-Instantiated Recurrent Units
# with Dynamic Soft Recursion

**Aston Zhang**[†]**, Yi Tay**[‡,*]**, Yikang Shen**[•]**, Alvin Chan**[◁]**, Shuai Zhang**[◇]
[†]Amazon Web Services AI, [‡]Google Research
[•]Mila, Université de Montréal, [◁]NTU, Singapore, [◇]ETH Zürich
az@astonzhang.com

## Abstract

While standard recurrent neural networks explicitly impose a chain structure on different forms of data, they do not have an explicit bias towards recursive self-instantiation where the extent of recursion is dynamic. Given diverse and even growing data modalities (e.g., logic, algorithmic input and output, music, code, images, and language) that can be expressed in sequences and may benefit from more architectural flexibility, we propose the self-instantiated recurrent unit (Self-IRU) with a novel inductive bias towards dynamic soft recursion. On one hand, the Self-IRU is characterized by recursive self-instantiation via its gating functions, i.e., gating mechanisms of the Self-IRU are controlled by instances of the Self-IRU itself, which are repeatedly invoked in a recursive fashion. On the other hand, the extent of the Self-IRU recursion is controlled by gates whose values are between 0 and 1 and may vary across the temporal dimension of sequences, enabling dynamic soft recursion depth at each time step. The architectural flexibility and effectiveness of our proposed approach are demonstrated across multiple data modalities. For example, the Self-IRU achieves state-of-the-art performance on the logical inference dataset [Bowman et al., 2014] even when comparing with competitive models that have access to ground-truth syntactic information.

## 1 Introduction

Models based on the notion of recurrence have enjoyed pervasive impact across various applications. In particular, most effective recurrent neural networks (RNNs) operate with gating functions. Such gating functions not only ameliorate vanishing gradient issues when modeling and capturing long-range dependencies, but also benefit from fine-grained control over temporal composition for sequences [Hochreiter and Schmidhuber, 1997, Cho et al., 2014].

With diverse and even growing data modalities (e.g., logic, algorithmic input and output, music, code, images, and language) that can be expressed in sequences and may benefit from more architectural flexibility, recurrent neural networks that only explicitly impose a chain structure on such data but lack an explicit bias towards recursive self-instantiation may be limiting. For example, their gating functions are typically static across the temporal dimension of sequences. In view of such, this paper aims at studying an inductive bias towards recursive self-instantiation where the extent of recursion is dynamic at different time steps.

We propose a novel recurrent unit whose gating functions are repeatedly controlled by instances of the recurrent unit itself. Our proposed model is called the self-instantiated recurrent unit (Self-IRU), where self-instantiation indicates modeling via the own instances of the model itself in a recursive fashion. Specifically, two gates of the Self-IRU are controlled by Self-IRU instances. Biologically,

---

[*]Work was done at NTU.

35th Conference on Neural Information Processing Systems (NeurIPS 2021).

this design is motivated by the prefrontal cortex/basal ganglia working memory indirection [Kriete et al., 2013]. For example, a child Self-IRU instance drives the gating for outputting from its parent Self-IRU instance.

Our proposed Self-IRU is also characterized by the dynamically controlled recursion depths. Specifically, we design a dynamic soft recursion mechanism, which softly learns the depth of recursive self-instantiation on a per-time-step basis. More concretely, certain gates are reserved to control the extent of the Self-IRU recursion. Since values of these gates are between 0 and 1 and may vary across the temporal dimension, they make dynamic soft recursion depth at each time step possible, which could lead to more architectural flexibility across diverse data modalities.

This design of the Self-IRU is mainly inspired by the adaptive computation time (ACT) [Graves, 2016] that learns the number of computational steps between an input and an output and recursive neural networks that operate on directed acyclic graphs. On one hand, the Self-IRU is reminiscent of the ACT, albeit operated at the parameter level. While seemingly similar, the Self-IRU and ACT are very different in the context of what the objective is. Specifically, the goal of the Self-IRU is to dynamically expand the parameters of the model, not dynamically decide how long to deliberate on input tokens in a sequence. On the other hand, the Self-IRU marries the benefit of recursive reasoning with recurrent models. However, in contrast to recursive neural networks, the Self-IRU is neither concerned with syntax-guided composition [Tai et al., 2015, Socher et al., 2013, Dyer et al., 2016, Wang and Pan, 2020] nor unsupervised grammar induction [Shen et al., 2017, Choi et al., 2018, Yogatama et al., 2016, Havrylov et al., 2019].

**Our Contributions** All in all, sequences are fundamentally native to the world, so the design of effective inductive biases for data in this form has far-reaching benefits across a diverse range of real-world applications. Our main contributions are outlined below:

- We propose the self-instantiated recurrent unit (Self-IRU). It is distinctly characterized by a novel inductive bias towards modeling via the own instances of the unit itself in a recursive fashion, where the extent of recursion is dynamically learned across the temporal dimension of sequences.

- We evaluate the Self-IRU on a wide spectrum of sequence modeling tasks across multiple modalities: logical inference, sorting, tree traversal, music modeling, semantic parsing, code generation, and pixel-wise sequential image classification. Overall, the empirical results demonstrate architectural flexibility and effectiveness of the Self-IRU. For example, the Self-IRU achieves state-of-the-art performance on the logical inference dataset [Bowman et al., 2014] even when comparing with competitive models that have access to ground-truth syntactic information.

**Notation** For readability, all vectors and matrices are denoted by lowercase and uppercase bold letters such as $\mathbf{x}$ and $\mathbf{X}$, respectively. When a scalar is added to a vector, the addition is applied element-wise [Zhang et al., 2021].

## 2 Method

This section introduces the proposed Self-IRU. The Self-IRU is fundamentally a recurrent model, but distinguishes itself in that the gating functions that control compositions over time are recursively modeled by instances of the Self-IRU itself, where the extent of recursion is dynamic. In the following, we begin with the model architecture that can recursively self-instantiate. Then we detail its key components such as how dynamic soft recursion is enabled.

### 2.1 Self-Instantiation

Given an input sequence of tokens $\mathbf{x}_1, \ldots, \mathbf{x}_T$, the Self-IRU transforms them into hidden states throughout all the time steps: $\mathbf{h}_1, \ldots, \mathbf{h}_T$. Denoting by $L$ the user-specified maximum recursion depth, the hidden state at time step $t$ is

$$\mathbf{h}_t = \text{Self-IRU}^{(L)}(\mathbf{x}_t, \mathbf{h}_{t-1}^{(L)}),$$

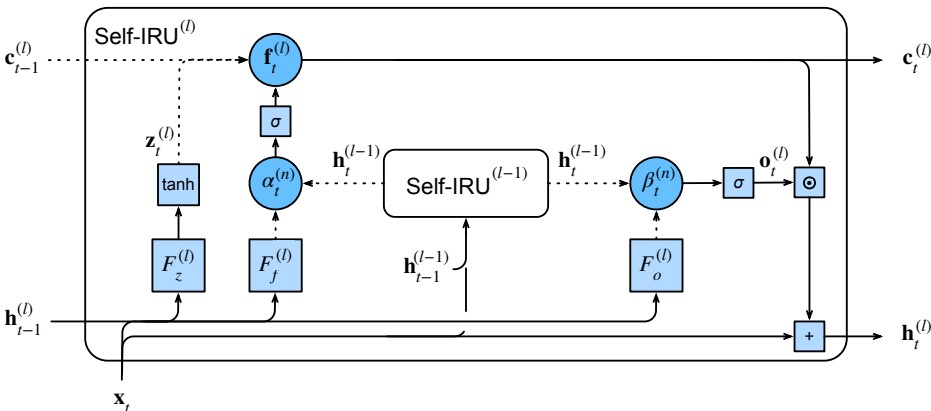

Figure 1: The self-instantiated recurrent unit (Self-IRU) model architecture. Circles represent gates that control information flow from dotted lines, and squares represent transformations or operators.

where Self-IRU$^{(L)}$ is an instance of the Self-IRU model at recursion depth $L$ and $\mathbf{h}_{t-1}^{(L)}$ is a hidden state at time step $t-1$ and recursion depth $L$. In general, a Self-IRU instance at any recursion depth $0 \leq l \leq L$ returns a hidden state for that depth:

$$\text{Self-IRU}^{(l)}(\mathbf{x}_t, \mathbf{h}_{t-1}^{(l)}) = \mathbf{h}_t^{(l)},$$

which involves the following computation:

$$\mathbf{f}_t^{(l)} = \sigma\left(\alpha_t^{(n)} \text{Self-IRU}^{(l-1)}(\mathbf{x}_t, \mathbf{h}_{t-1}^{(l-1)}) + (1 - \alpha_t^{(n)})F_f^{(l)}(\mathbf{x}_t, \mathbf{h}_{t-1}^{(l)})\right) \tag{2.1}$$

$$\mathbf{o}_t^{(l)} = \sigma\left(\beta_t^{(n)} \text{Self-IRU}^{(l-1)}(\mathbf{x}_t, \mathbf{h}_{t-1}^{(l-1)}) + (1 - \beta_t^{(n)})F_o^{(l)}(\mathbf{x}_t, \mathbf{h}_{t-1}^{(l)})\right) \tag{2.2}$$

$$\mathbf{z}_t^{(l)} = \tanh\left(F_z^{(l)}(\mathbf{x}_t, \mathbf{h}_{t-1}^{(l)})\right) \tag{2.3}$$

$$\mathbf{c}_t^{(l)} = \mathbf{f}_t^{(l)} \odot \mathbf{c}_{t-1}^{(l)} + (1 - \mathbf{f}_t^{(l)}) \odot \mathbf{z}_t^{(l)} \tag{2.4}$$

$$\mathbf{h}_t^{(l)} = \mathbf{o}_t^{(l)} \odot \mathbf{c}_t^{(l)} + \mathbf{x}_t, \tag{2.5}$$

where $\odot$ denotes the element-wise multiplication, $\sigma$ denotes the sigmoid function, scalars $\alpha_t^{(n)}$ and $\beta_t^{(n)}$ are soft depth gates at time step $t$ and recursion node $n$ in the unrolled recursion paths, and $F_f^{(l)}$, $F_o^{(l)}$, and $F_z^{(l)}$ are base transformations at recursion depth $l$. Without losing sight of the big picture, we will provide more details of such soft depth gates and base transformations later.

On a high level, Figure 1 depicts the Self-IRU model architecture. We highlight that two gating functions of a Self-IRU, the forget gate $\mathbf{f}_t^{(l)}$ in (2.1) and the output gate $\mathbf{o}_t^{(l)}$ in (2.2), are recursively controlled by instances of the Self-IRU itself. Therefore, we call both the forget and output gates the *self-instantiation gates*. The base case ($l = 0$) for self-instantiation gates is

$$\mathbf{f}_t^{(0)} = \sigma\left(F_f^{(0)}(\mathbf{x}_t, \mathbf{h}_{t-1}^{(0)})\right) \text{ and } \mathbf{o}_t^{(0)} = \sigma\left(F_o^{(0)}(\mathbf{x}_t, \mathbf{h}_{t-1}^{(0)})\right).$$

At each recursion depth $l$, the candidate memory cell $\mathbf{z}_t^{(l)}$ at time step $t$ is computed in (2.3). Then in (2.4), the forget gate $\mathbf{f}_t^{(l)}$ controls the information flow from $\mathbf{z}_t^{(l)}$ and the memory cell $\mathbf{c}_{t-1}^{(l)}$ at the previous time step to produce the memory cell $\mathbf{c}_t^{(l)}$ at the current time step $t$. As illustrated by the bottom arrow starting from $\mathbf{x}_t$ in Figure 1, the output gating in (2.5) also adds a skip connection from residual networks to facilitate gradient flow throughout the recursive self-instantiation of the Self-IRU [He et al., 2016].

## 2.2 Dynamic Soft Recursion

Now let us detail the soft depth gates $\alpha_t^{(n)}$ and $\beta_t^{(n)}$ in (2.1) and (2.2) for time step $t$ and recursion node $n$ in the unrolled recursion paths. The index $n$ is used to distinguish nodes at different positions

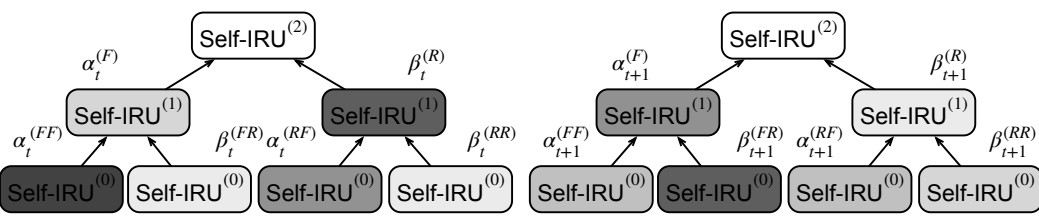

Figure 2: Soft depth gates $\alpha_t^{(n)}$ and $\beta_t^{(n)}$ for time step $t$ and recursion node $n$ (denoting $F$ and $R$ as the left child and the right child, respectively) control the extent of the Self-IRU recursion. The extent is indicated by greyscale of any node at the beginning of an arrow along an unrolled recursion path. These gates are between 0 and 1 and may vary across the temporal dimension, enabling dynamic soft recursion depth at each time step (here maximum depth $L = 2$).

in the recursion tree (e.g., in Figure 2) that is determined by the maximum recursion depth $L$. We propose learning them in a data-driven fashion. Specifically, we parameterize $\alpha_t^{(n)}$ and $\beta_t^{(n)}$ with

$$\alpha_t^{(n)} = \sigma(F_\alpha^{(n)}(\mathbf{x}_t)) \text{ and } \beta_t^{(n)} = \sigma(F_\beta^{(n)}(\mathbf{x}_t)),$$

where $F_*^{(n)}(\mathbf{x}_t) = \mathbf{W}_*^{(n)}\mathbf{x}_t + b_*^{(n)} \in \mathbb{R}$ ($* \in \{\alpha, \beta\}$) with weight parameters $\mathbf{W}_*^{(n)}$ and bias parameters $b_*^{(n)}$ both learned from data.

Together with the sigmoid function $\sigma$, these simple linear transformations of the input token $\mathbf{x}_t$ are applied dynamically at each time step $t$ across the input sequence. Moreover, as shown in (2.1) and (2.2), mathematically $0 < \alpha_t^{(n)}, \beta_t^{(n)} < 1$ control the extent of recursion at each recursion node $n$, enabling soft depth along any unrolled recursive self-instantiation path. Thus, $\alpha_t^{(n)}$ and $\beta_t^{(n)}$ are called the *soft depth gates*.

Putting these together, Figure 2 unrolls the recursive self-instantiation paths at two consecutive time steps to illustrate dynamic soft recursion depth. Specifically, the "softness" is indicated by greyscale of any node at the beginning of an arrow along an unrolled recursion path. In sharp contrast to multi-layer RNNs, Self-IRUs enable tree structures of self-instantiation, where the extent of recursion is dynamic (to be visualized in Section 3.7).

### 2.3 Base Transformations

At any recursion depth $l$, $F_f^{(l)}$ in (2.1), $F_o^{(l)}$ in (2.2), and $F_z^{(l)}$ in (2.3) are base transformations of the input $\mathbf{x}_t$ and the hidden state $\mathbf{h}_{t-1}^{(l)}$. For example, we can model base transformations using RNN units (e.g., LSTM): at recursion depth $l$, for $* \in \{f, o, z\}$ we have

$$F_*^{(l)}(\mathbf{x}_t, \mathbf{h}_{t-1}^{(l)}) = \text{RNN}_*^{(l)}(\mathbf{x}_t, \mathbf{h}_{t-1}^{(l)}).$$

Alternatively, we may also model base transformations with linear layers that only transform the input $\mathbf{x}_t$ using learnable weight parameters $\mathbf{W}_*^{(l)}$ and bias parameters $\boldsymbol{b}_*^{(l)}$ for $* \in \{f, o, z\}$:

$$F_*^{(l)}(\mathbf{x}_t) = \mathbf{W}_*^{(l)}\mathbf{x}_t + \boldsymbol{b}_*^{(l)}.$$

The Self-IRU is agnostic to the choice of base transformations and we will evaluate different choices in the experiments. We will discuss how the Self-IRU can be useful as a (parallel) non-autoregressive model and connects to other recurrent models in the supplementary material.

## 3 Experiments

To demonstrate the architectural flexibility and effectiveness, we evaluate Self-IRUs on a wide range of publicly available benchmarks, perform ablation studies on the maximum recursion depth and base transformations, and analyze dynamics of soft depth gates.

### 3.1 Pixel-wise Sequential Image Classification

The sequential pixel-wise image classification problem treats pixels in images as sequences. We use the well-established pixel-wise MNIST and CIFAR-10 datasets.

Table 1: Experimental results (accuracy) on the pixel-wise sequential image classification task.

| Model | #Params | MNIST | CIFAR-10 |
|---|---|---|---|
| Independently R-RNN [Li et al., 2018a] | - | 99.00 | - |
| r-LSTM with Aux Loss [Trinh et al., 2018] | - | 98.52 | 72.20 |
| Transformer (self-attention) [Trinh et al., 2018] | - | 98.90 | 62.20 |
| TrellisNet [Bai et al., 2018b] (reported) | 8.0M | **99.20** | **73.42** |
| TrellisNet [Bai et al., 2018b] (our run) | 8.0M | 97.59 | 55.83 |
| Self-IRU | 0.9M | 99.04 | 73.01 |

Table 1 reports the results of Self-IRUs against independently recurrent RNNs [Li et al., 2018a], r-LSTMs with aux loss [Trinh et al., 2018], Transformers (self-attention) [Trinh et al., 2018], and TrellisNets [Bai et al., 2018b]. On both the MNIST and CIFAR-10 datasets, the Self-IRU outperforms most of the other investigated baseline models. For the only exception, parameters of the Self-IRU are only about $1/8$ of those of the TrellisNet [Bai et al., 2018b] while still achieving comparable performance. This supports that the Self-IRU is a reasonably competitive sequence encoder.

### 3.2 Logical Inference

We experiment for the logical inference task on the standard dataset[2] proposed by Bowman et al. [2014]. This classification task is to determine the semantic equivalence of two statements expressed with logic operators such as *not*, *and*, and *or*. As per prior work [Shen et al., 2018], the model is trained on sequences with 6 or fewer operations and evaluated on sequences of 6 to 12 operations.

Table 2: Experimental results (accuracy) on the logical inference task (symbol † denotes models with access to ground-truth syntax). The baseline results are reported from [Shen et al., 2018]. The Self-IRU achieves state-of-the-art performance.

| Model | #Operations | | | | | |
|---|---|---|---|---|---|---|
| | $= 7$ | $= 8$ | $= 9$ | $= 10$ | $= 11$ | $= 12$ |
| Tree-LSTM† [Tai et al., 2015] | 93.0 | 90.0 | 87.0 | 89.0 | 86.0 | 87.0 |
| LSTM [Bowman et al., 2014] | 88.0 | 85.0 | 80.0 | 78.0 | 71.0 | 69.0 |
| RRNet [Jacob et al., 2018] | 84.0 | 81.0 | 78.0 | 74.0 | 72.0 | 71.0 |
| ON-LSTM [Shen et al., 2018] | 91.0 | 87.0 | 86.0 | 81.0 | 78.0 | 76.0 |
| Self-IRU | **97.0** | **95.0** | **93.0** | **92.0** | **90.0** | **88.0** |

We compare Self-IRUs with Tree-LSTMs [Tai et al., 2015], LSTMs [Bowman et al., 2014], RR-Nets [Jacob et al., 2018], and ordered-neuron (ON-) LSTMs [Shen et al., 2018] based on the common experimental setting in these works. Table 2 reports our results on the logical inference task. The Self-IRU is a strong and competitive model on this task, outperforming ON-LSTM by a wide margin ($+12\%$ on the longest number of operations). Notably, the Self-IRU achieves state-of-the-art performance on this dataset even when comparing with Tree-LSTMs that have access to ground-truth syntactic information.

### 3.3 Sorting and Tree Traversal

We also evaluate Self-IRUs on two algorithmic tasks that are solvable by recursion: sorting and tree traversal. In sorting, the input to the model is a sequence of integers. The correct output is the sorted sequence of integers. Since mapping sorted inputs to outputs can be implemented in a recursive fashion, we evaluate the Self-IRU's ability to model recursively structured sequence data. An example input-output pair would be $9, 1, 10, 5, 3 \rightarrow 1, 3, 5, 9, 10$. We evaluate on sequence length $m = \{5, 10\}$.

---

[2] https://github.com/sleepinyourhat/vector-entailment.

In the tree traversal problem, we construct a binary tree of maximum depth $n$. The goal is to generate the postorder tree traversal given the inorder and preorder traversal of the tree. This is known to arrive at only one unique solution. The constructed trees have random sparsity where trees can grow up to maximum depth $n$. Hence, these trees can have varying depths (models can solve the task entirely when trees are fixed and full). We concatenate the postorder and inorder sequences, delimited by a special token. We evaluate on maximum depth $n \in \{3, 4, 5, 8, 10\}$. For $n \in \{5, 8\}$, we ensure that each tree traversal has at least 10 tokens. For $n = 10$, we ensure that each path has at least 15 tokens. An example input-output pair would be $13, 15, 4, 7, 5, X, 13, 4, 15, 5, 7 \rightarrow 7, 15, 13, 4, 5$.

We frame sorting and tree traversal as sequence-to-sequence [Sutskever et al., 2014] tasks and evaluate models with measures of exact match (EM) accuracy and perplexity (PPL). We use a standard encoder-decoder architecture with attention [Bahdanau et al., 2014], and vary the encoder module with BiLSTMs, stacked BiLSTMs, and ordered-neuron (ON-) LSTMs [Shen et al., 2018].

Table 3: Experimental results on the sorting and tree traversal tasks.

| | Sorting | | | | Tree Traversal | | | | | | | | | |
| | $m = 5$ | | $m = 10$ | | $n = 3$ | | $n = 4$ | | $n = 5$ | | $n = 8$ | | $n = 10$ | |
| Model | EM | PPL | EM | PPL | EM | PPL | EM | PPL | EM | PPL | EM | PPL | EM | PPL |
|---|---|---|---|---|---|---|---|---|---|---|---|---|---|---|
| BiLSTM | 79.9 | 1.2 | 78.9 | 1.2 | 100 | 1.0 | 96.9 | 2.4 | 60.3 | 2.4 | 5.6 | 30.6 | 2.2 | 132.0 |
| Stacked BiLSTM | 83.4 | 1.2 | 88.0 | 1.1 | 100 | 1.0 | 98.0 | 1.0 | 63.4 | 2.5 | **5.9** | 99.9 | 2.8 | 225.1 |
| ON-LSTM | 90.8 | 1.1 | 87.4 | 1.1 | 100 | 1.0 | 81.0 | 1.4 | 55.7 | 2.8 | 5.5 | 52.3 | 2.7 | 173.2 |
| Self-IRU | **92.2** | **1.1** | **90.6** | **1.1** | 100 | 1.0 | **98.4** | **1.0** | **63.4** | **1.8** | 5.6 | **20.4** | **2.8** | **119.0** |

Table 3 reports our results on the sorting and tree traversal tasks. In fact, all the models solve the tree traversal task with $n = 3$. However, the task gets increasingly harder with a greater maximum possible depth and largely still remains a challenge for neural models today. On one hand, stacked BiLSTMs always perform better than BiLSTMs and ON-LSTMs occasionally perform worse than standard BiLSTMs on tree traversal, while for the sorting task ON-LSTMs perform much better than standard BiLSTMs. On the other hand, the relative performance of the Self-IRU is generally better than any of these baselines, especially pertaining to perplexity.

## 3.4  Music Modeling

Moreover, we evaluate the Self-IRU on the polyphonic music modeling task, i.e., generative modeling of musical sequences. We use three well-established datasets: Nottingham, JSB Chorales, and Piano Midi [Boulanger-Lewandowski et al., 2012]. The inputs are 88-bit (88 piano keys) sequences.

Table 4: Experimental results (negative log-likelihood) on the music modeling task.

| Model | Nottingham | JSB | Piano Midi |
|---|---|---|---|
| GRU [Chung et al., 2014] | 3.13 | 8.54 | 8.82 |
| LSTM [Song et al., 2019] | 3.25 | 8.61 | 7.99 |
| G2-LSTM [Li et al., 2018b] | 3.21 | 8.67 | 8.18 |
| B-LSTM [Song et al., 2019] | 3.16 | 8.30 | 7.55 |
| TCN [Bai et al., 2018a] (reported) | 3.07 | **8.10** | - |
| TCN [Bai et al., 2018a] (our run) | 2.95 | 8.13 | 7.53 |
| Self-IRU | **2.88** | 8.12 | **7.49** |

Table 4 compares the Self-IRU with a wide range of published works: GRU [Chung et al., 2014], LSTM [Song et al., 2019], G2-LSTM [Li et al., 2018b], B-LSTM [Song et al., 2019], and TCN [Bai et al., 2018a]. The Self-IRU achieves the best performance on the Nottingham and Piano midi datasets. It also achieves competitive performance on the JSB Chorales dataset, only underperforming the state-of-the-art by 0.02 negative log-likelihood.

## 3.5  Semantic Parsing and Code Generation

We further evaluate Self-IRUs on the semantic parsing (the Geo, Atis, and Jobs datasets) and code generation (the Django dataset) tasks. They are mainly concerned with learning to parse and generate structured data. We run our experiments on the publicly released source code[3] of [Yin and Neubig,

---

[3] https://github.com/pcyin/tranX

2018], replacing the recurrent decoder with our Self-IRU decoder (TranX + Self-IRU). We only replace the recurrent decoder since our early experiments showed that varying the encoder did not yield any benefits in performance. Overall, our hyperparameter details strictly follow the codebase of [Yin and Neubig, 2018], i.e., we run every model from their codebase as it is.

Table 5: Experimental results (accuracy) on the semantic parsing (the Geo, Atis, and Jobs datasets) and code generation tasks (the Django dataset).

| Model | Geo | Atis | Jobs | Django |
|---|---|---|---|---|
| Seq2Tree [Dong and Lapata, 2016] | 87.1 | 84.6 | - | 31.5 |
| LPN [Ling et al., 2016] | - | - | - | 62.3 |
| NMT [Neubig, 2015] | - | - | - | 45.1 |
| YN17 [Yin and Neubig, 2017] | - | - | - | 71.6 |
| ASN [Rabinovich et al., 2017] | 85.7 | 85.3 | - | - |
| ASN + Supv. Attn. [Rabinovich et al., 2017] | 87.1 | 85.9 | - | - |
| TranX [Yin and Neubig, 2018] (reported in code) | **88.6** | 87.7 | 90.0 | 77.2 |
| TranX [Yin and Neubig, 2018] (our run) | 87.5 | 87.5 | 90.0 | 76.7 |
| TranX + Self-IRU | **88.6** | **88.4** | **90.7** | **78.3** |

Table 5 reports the experimental results in comparison with the other competitive baselines such as Seq2Tree [Dong and Lapata, 2016], LPN [Ling et al., 2016], NMT [Neubig, 2015], YN17 [Yin and Neubig, 2017], ASN (with and without supervised attention) [Rabinovich et al., 2017], and TranX [Yin and Neubig, 2018]. We observe that TranX + Self-IRU outperforms all the other approaches, achieving state-of-the-art performance. On the code generation task, TranX + Self-IRU outperforms TranX by +1.6% and ≈ +1% on all the semantic parsing tasks. More importantly, the performance gain over the base TranX method allows us to observe the ablative benefit of the Self-IRU that is achieved by only varying the recurrent decoder.

## 3.6 Ablation Studies of the Maximum Recursion Depth and Base Transformations

Table 6 presents ablation studies of the maximum recursion depth (Section 2.1) and base transformations (Section 2.3) of Self-IRUs. The results are based on the semantic parsing (Atis) and code generation (Django) tasks. We can see that their optimal choice is task dependent: (i) on the semantic parsing task, using the linear layer performs better than the LSTM for base transformations; (ii) conversely, the linear transformation performs worse than the LSTM on the code generation task.

Table 6: Ablation studies of the maximum recursion depth and base transformation on the semantic parsing (SP) and code generation (CG) tasks.

| Max Depth | Base Transformations | SP | CG |
|---|---|---|---|
| 1 | Linear | **88.40** | 77.56 |
| 2 | Linear | 88.21 | 77.62 |
| 3 | Linear | 87.80 | 76.84 |
| 1 | LSTM | 86.61 | **78.33** |
| 2 | LSTM | 85.93 | 77.39 |

On the whole, we also observe this across the other tasks in the experiments. Table 7 reports their optimal combinations for diverse tasks in the experiments, where the maximum recursion depth is evaluated on $L = \{0, 1, 2, 3\}$. As we can tell from different optimal combinations in Table 7, choices of the maximum recursion depth and base transformations of Self-IRUs depend on tasks.

Table 7: The optimal maximum recursion depth and base transformations for different tasks in the experiments.

| Task | Max Depth | Base Transformations |
|---|---|---|
| Image classification | 1 | LSTM |
| Logical inference | 2 | LSTM |
| Tree traversal | 1 | LSTM |
| Sorting | 1 | LSTM |
| Music modeling | 2 | Linear |
| Semantic parsing | 1 | Linear |
| Code generation | 1 | LSTM |

## 3.7 Analysis of Soft Depth Gates

Besides the task-specific maximum recursion depth and base transformations, empirical effectiveness of Self-IRUs may also be explained by the modeling flexibility via the inductive bias towards dynamic soft recursion (Section 2.2). We will analyze in two aspects below.

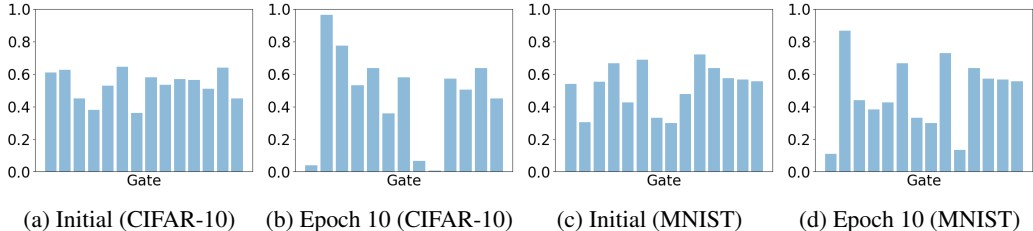

(a) Initial (CIFAR-10)    (b) Epoch 10 (CIFAR-10)    (c) Initial (MNIST)    (d) Epoch 10 (MNIST)

Figure 3: Soft depth gate values at initialization and training epoch 10 on the CIFAR-10 and MNIST datasets.

First, during training, the Self-IRU has the flexibility of building data-dependent recursive patterns of self-instantiation. Figure 3 displays values of all the soft depth gates at all the three recursion depths on the CIFAR-10 and MNIST datasets, depicting how the recursive pattern of the Self-IRU is updated during training. For different datasets, the Self-IRU also flexibly learns to construct different soft recursive (via soft depth gates of values between 0 and 1) patterns.

Second, we want to find out whether the Self-IRU has the flexibility of softly learning the recursion depth on a per-time-step basis via the inductive bias towards dynamic soft recursion. Figure 4 visualizes such patterns (i) for pixel-wise sequential image classification on the CIFAR-10 and MNIST datasets and (ii) for music modeling on the Nottingham dataset. Notably, all the datasets have very diverse temporal compositions of recursive patterns. More concretely, the soft depth gate values fluctuate aggressively on the CIFAR-10 dataset (consisting of color images) in Figure 4a while remaining more stable for music modeling in Figure 4c. Moreover, these soft depth gate values remain totally constant on the MNIST dataset (consisting of much simpler grayscale images) in Figure 4b. These provide compelling empirical evidence for the architectural flexibility of Self-IRUs: they can adjust the dynamic construction adaptively and can even revert to static recursion over time if necessary (such as for simpler tasks).

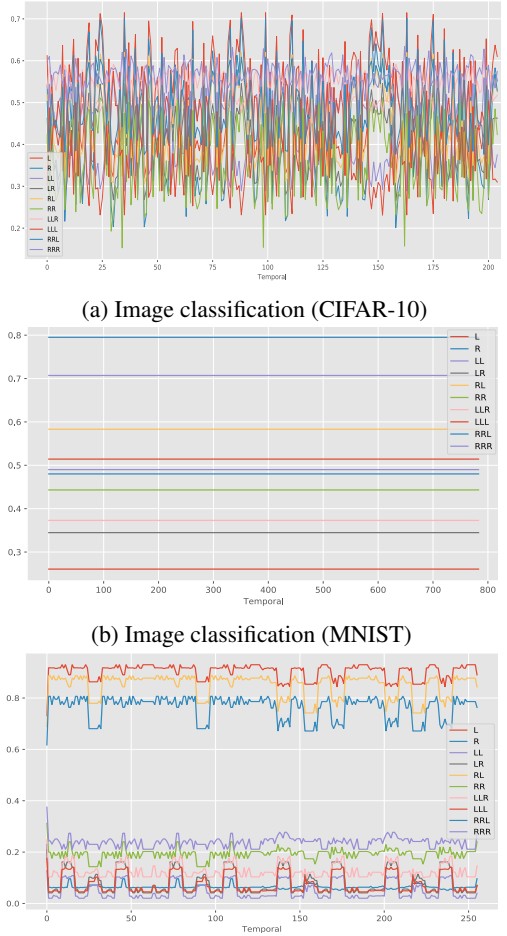

(a) Image classification (CIFAR-10)

(b) Image classification (MNIST)

(c) Music modeling (Nottingham)

Figure 4: Soft depth gate values across the temporal dimension. "L" and "R" denote $\alpha_t^{(n)}$ and $\beta_t^{(n)}$, respectively (e.g., "LLR" denotes the node at the end of the unrolled recursive path $\alpha_t^{(n)} \rightarrow \alpha_t^{(n)} \rightarrow \beta_t^{(n)}$).

The dynamic soft recursion pattern is made more intriguing by observing how the "softness" alters on the CIFAR-10 and Nottingham datasets. From Figure 4c we observe that the soft recursion pattern of the model changes in a rhythmic fashion, in line with our intuition of musical data. When dealing with pixel information, the recursive pattern in Figure 4a changes adaptively according to the more complex color-image information. Though these empirical results are intuitive, a better understanding of such behaviors may benefit from theoretical or biological perspectives in the future.

## 4   Related Work

The study of effective inductive biases for sequential representation learning has been a prosperous research direction. This has spurred on research across multiple fronts, starting from gated recurrent models [Hochreiter and Schmidhuber, 1997, Cho et al., 2014], convolution [Kim, 2014], to self-attention-based models [Vaswani et al., 2017].

The intrinsic hierarchical structure native to many forms of sequences has long fascinated and inspired researchers [Socher et al., 2013, Bowman et al., 2014, 2016, Dyer et al., 2016]. Nested LSTMs use hierarchical memories [Moniz and Krueger, 2017]. The study of recursive networks, popularized by Socher et al. [2013], has provided a foundation for learning syntax-guided composition. Along the same vein, Tai et al. [2015] proposed Tree-LSTMs that guide LSTM composition with grammar. Recent attempts have been made to learn this process without guidance or syntax-based supervision [Yogatama et al., 2016, Shen et al., 2017, Choi et al., 2018, Havrylov et al., 2019, Kim et al., 2019]. Specifically, ordered-neuron LSTMs [Shen et al., 2018] propose structured gating mechanisms, imbuing the recurrent unit with a tree-structured inductive bias. Besides, Tran et al. [2018] showed that recurrence is important for modeling hierarchical structure. Notably, learning hierarchical representations across multiple time-scales [El Hihi and Bengio, 1996, Schmidhuber, 1992, Koutnik et al., 2014, Chung et al., 2016, Hafner et al., 2017] has also demonstrated effectiveness.

Learning an abstraction and controller over a base recurrent unit is also another compelling direction. First proposed in fast weights by Schmidhuber [1992], several recent works explored this notion. HyperNetworks [Ha et al., 2016] learn to generate weights for another recurrent unit, i.e., a form of relaxed weight sharing. On the other hand, RCRN [Tay et al., 2018] explicitly parameterizes the gates of an RNN unit with other RNN units. Recent studies on the recurrent unit are also reminiscent of this particular notion [Bradbury et al., 2016, Lei et al., 2018].

The fusion of recursive and recurrent architectures is also notable. This direction is probably the closest relevance to our proposed method, although with vast differences. Liu et al. [2014] proposed recursive recurrent networks for machine translation that are concerned with the more traditional syntactic supervision concept of vanilla recursive networks. Jacob et al. [2018] proposed the RRNet, which learns hierarchical structures on the fly. The RRNet proposes to learn to split or merge nodes at each time step, which makes it reminiscent of other works [Choi et al., 2018, Shen et al., 2018]. Lee and Osindero [2016] and Aydin and Güngör [2020] proposed to feed recursive neural network output into recurrent models. Alvarez-Melis and Jaakkola [2016] proposed doubly recurrent decoders for tree-structured decoding. The core of their method is a depth and breath-wise recurrence which is similar to our model. However, our Self-IRU is concerned with learning recursive self-instantiation, which is in sharp contrast to their objective of decoding trees.

Last, our work combines the idea of external meta-controllers [Schmidhuber, 1992, Ha et al., 2016, Tay et al., 2018] with recursive architectures. Specifically, our recursive parameterization is also a form of dynamic memory that offers improved expressiveness in similar spirit to memory-augmented recurrent models [Santoro et al., 2018, Graves et al., 2014, Tran et al., 2016].

## 5   Summary and Discussions

We proposed the Self-IRU that is characterized by recursive instantiation of the model itself, where the extent of the recursion may vary temporally. The experiments across multiple modalities demonstrated the architectural flexibility and effectiveness of the Self-IRU. While there is a risk of abusing the Self-IRU such as for generating fake contents, we believe that our model is overall beneficial through effective understanding of our digitalized world across diverse modalities.

**Acknowledgements.**   We thank the anonymous reviewers for the insightful comments on this paper.

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
