# A  Appendix

## A.1  Non-Autoregressive Variant

We postulate that the Self-IRU can also be useful as a (parallel) non-autoregressive model. This can be interpreted as a form of recursive feed-forward layers that is used in place of the recurrent and autoregressive Self-IRU.

By dispensing with the reliance on the hidden states of the previous time steps and the memory cells, the non-autoregressive variant of the Self-IRU is redefined as

$$\mathbf{f}_t^{(l)} = \sigma\Big(\alpha_t^{(n)} \, \text{Self-IRU}^{(l-1)}(\mathbf{x}_t) + (1 - \alpha_t^{(n)})F_f^{(l)}(\mathbf{x}_t)\Big)$$

$$\mathbf{o}_t^{(l)} = \sigma\Big(\beta_t^{(n)} \, \text{Self-IRU}^{(l-1)}(\mathbf{x}_t) + (1 - \beta_t^{(n)})F_o^{(l)}(\mathbf{x}_t)\Big)$$

$$\mathbf{z}_t^{(l)} = \tanh\Big(F_z^{(l)}(\mathbf{x}_t)\Big)$$

$$\widetilde{\mathbf{h}}_t^{(l)} = \mathbf{f}_t^{(l)} \odot \mathbf{x}_t + \mathbf{o}_t^{(l)} \odot \mathbf{z}_t^{(l)}$$

$$\mathbf{h}_t^{(l)} = \widetilde{\mathbf{h}}_t^{(l)} + \mathbf{x}_t.$$

As we will show in the experiments, this non-autoregressive Self-IRU variant can be used in place of position-wise feed-forward layers in the Transformer architecture [Vaswani et al., 2017]. In this case, base transformations $F_*^{(l)}$ for $* \in \{f, o, z\}$ are position-wise functions as well.

## A.2  Experiments on Neural Machine Translation

To evaluate whether the non-autoregressive variant of the Self-IRU can boost the performance of Transformers, we conduct neural machine translation experiments on two IWSLT datasets that are collections derived from TED talks. Specifically, we compare on the IWSLT 2014 German-English (De-En) and IWSLT 2015 English-Vietnamese (En-Vi) datasets.

We replace the multi-head aggregation layer in the Transformer [Vaswani et al., 2017] with the non-autoregressive variant of the Self-IRU as described in Section A.1, setting base transformations with linear layers and the maximum recursion depth as 2. For our experiments, we use the standard implementation and hyperparameters in Tensor2Tensor[4] [Vaswani et al., 2018] with the base setting for Transformers. For evaluation, model averaging is used (latest 5 checkpoints) and the beam search with beam size 8 (De-En) and 4 (En-Vi) and length penalty of 0.6 is adopted for decoding.

Table 8: Experimental results (BLEU) on the IWSLT 2014 De-En and IWSLT 2015 En-Vi neural machine translation tasks.

| Model | IWSLT 2014 De-En | IWSLT 2015 En-Vi |
|---|---|---|
| Transformer [Vaswani et al., 2017] | 36.30 | 28.43 |
| Transformer + Self-IRU | **37.09** | **30.81** |

Table 8 reports results on the two neural machine translation tasks. On the IWSLT 2014 De-En and IWSLT 2015 En-Vi datasets, the non-autoregressive Self-IRU variant boosts the performance of the Transformer by $+0.79$ and $+2.38$ BLEU, respectively.

## A.3  Connections to Other Recurrent Models

Self-IRUs can be considered as a broader framework. Here we briefly discuss how Self-IRUs connect to some other recurrent models.

For example, according to the base case of the self-instantiation gates, when setting the maximum recursive depth $L = 0$, the Self-IRU can be reverted to a simple recurrent unit (SRU) [Lei et al.,

---

[4]`https://github.com/tensorflow/tensor2tensor`

2018][5]:

$$\mathbf{f}_t = \sigma\Big(F_f(\mathbf{x}_t, \mathbf{h}_{t-1})\Big)$$

$$\mathbf{o}_t = \sigma\Big(F_o(\mathbf{x}_t, \mathbf{h}_{t-1})\Big)$$

$$\mathbf{z}_t = \tanh\Big(F_z(\mathbf{x}_t, \mathbf{h}_{t-1})\Big)$$

$$\mathbf{c}_t = \mathbf{f}_t \odot \mathbf{c}_{t-1} + (1 - \mathbf{f}_t) \odot \mathbf{z}_t$$

$$\mathbf{h}_t = \mathbf{o}_t \odot \mathbf{c}_t + \mathbf{x}_t.$$

However, the key difference between SRUs and standard RNNs is that the previous hidden state $\mathbf{h}_{t-1}$ is not used in the base transformations $F_f$, $F_o$, and $F_z$.

Furthermore, when setting the base transformation $F_*(\mathbf{x}_t, \mathbf{h}_{t-1}) = \mathbf{W}_*([\mathbf{x}_t; \mathbf{h}_{t-1}]) + \boldsymbol{b}_*$ for $* \in \{f, o, z\}$, where $(;)$ denotes concatenation, the Self-IRU is similar to standard gated recurrent models such as LSTMs and GRUs.

Similarly, when $L = 0$ and $F_*$ for $* \in \{f, o, z\}$ is a one-dimensional convolutional model, the Self-IRU becomes the Quasi RNN model [Bradbury et al., 2016]. Alternatively, when $L = 0$ and $F_*$ for $* \in \{f, o, z\}$ is an LSTM or a GRU unit, the Self-IRU takes the form of the recurrently controlled recurrent network model [Tay et al., 2018].

---

[5]The original paper uses the skip connection from highway networks [Srivastava et al., 2015].