# OpenReview forum: "Self-Instantiated Recurrent Units with Dynamic Soft Recursion"
_NeurIPS.cc/2021/Conference — NeurIPS 2021 Poster_

### Official Review · Reviewer_yiWz · 2021-07-16

**Rating:** 7
**Confidence:** 4

**Summary:**

This paper proposes a novel gated recurrent unit module that accomplishes a form of soft recursion through dynamic gating. The experiments evaluate the module across a range of tasks, and in particular show promising results on a logical inference task.

**Ethical Concerns:**

None.

**Limitations And Societal Impact:**

None.

**Main Review:**

This paper proposes a clever way to accomplish a form of soft recursion, by introducing a recurrent module, the 'self instantiated recurrent unit' (self-IRU), that can gate other instances of itself. I found the idea to be well motivated and the experiments seem to demonstrate improvements across a range of domains (though see the caveat below about statistical significance and multiple seeds), particularly on a logical inference task.

I found some parts of the proposed module difficult to understand, and hopefully this can be clarified in the revision. In particular, what is the relation of index $n$ (for the depth gates) to the index $l$ (for the depth layer)? The paper states that $l$ takes on values from $1$ to $L$, the maximum allowed depth (a hyperaparameter), but what values do $n$ take on, and how does that relate to the recursion depth? It may help to clarify this if the values of $n$ and $l$ are labelled in Figure 2, i.e. node 0 and layer 0 could be labelled with superscripts $(n=0)$ and $(l=0)$.

I'm also curious about a few of the design decisions, and am wondering whether the authors considered any of the following variations, or can offer any insight into the motivation behind these particular decisions:
* Rather than having the input to $z_{t}^{(l)}$ come from the original input $x_{t}$, instead basing this on the output of the Self-IRU in the previous layer, as is done for the forget and output gates.
* Sharing parameters for the base transformations $F$ across layers.
* Computing the depth gates ($\alpha$ and $\beta$) using a recurrent layer instead of a single feedfoward layer.

I did have two concerns. These concerns are the primary reason for my score, which I would be happy to raise if they are addressed:

1) Are the results shown for a single random seed only, or averaged over multiple seeds? If a single seed, was only one seed used, or are these results for the best seed, and if so out of how many? I think the ideal approach would be to average over multiple seeds and present some measure of error, but at the very least this should be clarified. On some of the datasets the improvements were very small and it's not clear if they are even statistically significant.

2) This seems somewhat unlikely to me, but I'm curious to what extent the results can be explained just by additional depth. How would the proposed approach compare to a multilayer LSTM with the same number of layers as a Self-IRU? One reason for thinking that this might partially explain the results is that the stacked BiLSTM performs almost identically to the Self-IRU on the tree traversal task.

I found the results on the logical inference task very impressive. It makes a lot of sense that an inductive bias for recursion would be helpful for such a task. Have the authors considered visualizing the model's operation on a few examples from this task? That seems like it might be an intuitive way to convey how the model operates, and why it's useful for such tasks.

Minor points:
* It would be good to include more detail about hyperparameters, including the optimizer, learning rate, initialization etc. This could go in the Appendix.
* I'm curious about what the relation might be, if any, to the following paper [Kriete et al.], in which one layer of a network is used to gate the output of another layer, and therefore accomplishes a form of indirection. I wonder if something similar could happen with the Self-IRU.

Kriete, T., Noelle, D. C., Cohen, J. D., & O’Reilly, R. C. (2013). Indirection and symbol-like processing in the prefrontal cortex and basal ganglia. Proceedings of the National Academy of Sciences, 110(41), 16390-16395.

**Update and final summary**

This paper proposes a novel recurrent gated unit that implements a soft form of recursion. There is a thorough empirical evaluation across a range of tasks, showing that the proposed Self-IRU generally results in improvements. Though the improvements are small on some tasks, the improvements on the logical inference task are particularly impressive, especially against a baseline that has access to ground-truth syntactic information.

**Time Spent Reviewing:**

4

---

> ### Author Response · Authors · 2021-08-10
> **Response to Reviewer yiWz**
>
> Thank you for the positive assessment, insightful comments, and being specific in questions for increasing your score.
>
>
> ## On the index $n$
>
> The index $n$ is used to distinguish nodes at different positions in the recursion tree that is determined by the max recursion depth $L$. For example, denoting "F" and "R" as the left child and the right child, the indices $n$ in the left tree of Figure 2 are
>
>
>    $\ \ \ \ \ \ \boldsymbol{\alpha}_t^{(n=F)} \ \ \ \ \  \ \ \ \ \ \ \ \ \ \ \ \ \ \ \  \boldsymbol{\beta}_t^{(n=R)}$
>
>     /     \        /     \
>
> $\boldsymbol{\alpha}_t^{(n=FF)}$ $\boldsymbol{\beta}_t^{(n=FR)}$  $\boldsymbol{\alpha}_t^{(n=RF)}$ $\boldsymbol{\beta}_t^{(n=RR)}$
>
>
> The indices $n$ in the right tree of Figure 2 are the same as above: only subscripts $t$ are replaced by $t+1$.
>
> Conceptually, each value of $n$ has its own parameterization in line 104. This will allow the Self-IRU to produce different soft depth gate values $\boldsymbol{\alpha}_t^{(n)}$ and $\boldsymbol{\beta}_t^{(n)}$ at different node positions for the same $t$. Note that such node-position-based parameterizations are shared across the temporal dimension of sequences: when $t$ changes, dynamic $\mathbf{x}_t$ will allow the Self-IRU to produce dynamic soft depth gate values (line 104). We will clarify this in Figure 2 and Section 2.2 in the revision.
>
>
>
> ## On considerations of design variations (3 bullet points)
>
> These are all very brilliant considerations. Here are our point-to-point responses:
>
> * For the three components under considerations: (i) the forget gate $\mathbf{f}_t^{(l)}$; (ii) the output gate $\mathbf{o}_t^{(l)}$; (iii) the candidate memory cell $\mathbf{z}_t^{(l)}$, we had performed early proof-of-concept experiments to assess different choices (including combinations) of these three components for introducing the dynamic soft recursion design, showing that the best results are always obtained via choosing both (i) and (ii). We think this is reasonable: the challenge of long-term information preservation and short-term input skipping in latent variable models is addressed by LSTMs, via the control using both the forget gate and the output gate; thus, increasing architectural flexibility in these two critical components (see analysis in Section 3.7) of LSTMs may lead to higher effectiveness, which is demonstrated in the experiments.
>
> * Our early proof-of-concept experiments had shown that sharing parameters of base transformations across recursion depths harms the performance. This aligns with a typical deep RNN design where each layer has its own parameterization: see (9.3.1) of https://d2l.ai/chapter_recurrent-modern/deep-rnn.html.
>
> * While this may increase computations, we would like to leave a full exploration of this direction to future work.
>
>
>
> ## On random seeds
>
> The results are averaged over five random seeds. See one example of how to specify it at our anonymous link: https://github.com/anonymous-sc/Self-IRU/blob/main/SelfIRU/poly_music/music_test.py#L37 We will provide more clarifications in the revision.
>
>
> ## On stacked BiLSTM's performance in the tree traversal task
>
> We would like to clarify that on the tree traversal task, (i) all the models solve the task with $n=3$; (ii) when $n=4, 5, 6, 7$, the overall performance of the Self-IRU is still the best, at least quite clear from its lowest perplexity (PPL) measure (Table 3). Note that only the best results are reported for the compared methods, where the stacked BiLSTM uses 3 layers and the Self-IRU has a max depth of 2. In other words, if the stacked BiLSTM used 2 layers, its results are even worse.
>
>
> ## On visualizing the model's operation
>
> We had considered this but found it not straightforward to map the model behavior along the sequence to steps of solving the problem. Instead, we visualized the model behavior in Section 3.7 to demonstrate the architectural flexibility in practice, which may explain the empirical effectiveness.
>
>
> ## On minor points of hyperparameter details
>
> Thank you for the suggestions. We will provide more hyperparameter details in the revision.
>
>
> ## On minor points of the paper "Indirection and symbol-like processing in the prefrontal cortex and basal ganglia"
>
> What a pleasant surprise.
>
> Let's consider Fig. 1 of https://www.pnas.org/content/110/41/16390. The upper-left box (with "Stripe1", "Stripe2", "Stripe3") acts like a Self-IRU node that is closest to the input and the upper-right box (with "Agent", "Verb", "Patient") resembles its child Self-IRU node in the recursion. In the prefrontal cortex/basal ganglia working memory indirection, the upper-right box drives the gating for outputting from the upper-left box. Similarly, the child Self-IRU node drives the gating for outputting from its parent Self-IRU node. Although there are also differences, we are more than happy to include a brief (hopefully thought-provoking) discussion of this possible biological motivation in the revision. We will thank you (anonymous reviewer) for helping us make the paper better in the acknowledgement.

---

> > ### Comment · Reviewer_yiWz · 2021-08-11
> > **Reply to the authors**
> >
> > Thanks to the authors for the thorough reply. Here are my point-by-point responses:
> >
> > **Clarification of algorithm**
> >
> > Thank you for the clarification. Just to make sure I understand, this means that there will actually be different versions of each Self-IRU$^{l}$, depending on the $\alpha$ and $\beta$ nodes that feed in to it? So in Figure 2, the different instances of Self-IRU$^{(1)}$ have different states, even for the same time point $t$, because of the distinct $\alpha$ and $\beta$ values controlling their forget and output gates? It may be helpful to clarify that both in the figure and in the specification of the algorithm. It might be better to start with figure 2 first, with a more thorough specification of the indices, and also perhaps illustrating the dependency on $x_{t}$ (showing that each instance/layer of Self-IRUs receive the same input $x_{t}$, but have different gate values dependent on their position in the hierarchy).
> >
> > **Design considerations**
> >
> > Thanks for the thorough response to these questions. So when the candidate memory cell $z_{t}$ was also based on the input from the previous layer (as the forget and output gates are in the current version), that either hurt performance or didn't yield any improvement?
> >
> > **Random seeds and statistical significance**
> >
> > Thanks for this clarification. It would be good to also to report the standard error to determine in which cases the improvements from the proposed method are statistically significant, though I know this may not be possible when some baseline results come from other papers that didn't report this.
> >
> > **Extent to which results can be explained by increased depth**
> >
> > I see now that the Self-IRU always outperforms the stacked BiLSTM on either the accuracy or perplexity measure. Here too it would be good to have a measure of error since some of these differences are quite small. But I appreciate the point that the BiLSTM actually has more layers than the Self-IRU in this case, making my broader concern (that the results might be explained by mere depth) unlikely.
> >
> > **Relationship to indirection in prefrontal cortex / basal ganglia**
> >
> > That's very interesting! Thanks for the illuminating discussion.

---

> > > ### Author Response · Authors · 2021-08-12
> > > **Further Response to Reviewer yiWz**
> > >
> > > Thank you for your fast and detailed reply. Below are our further point-by-point responses:
> > >
> > >
> > > **Clarification of algorithm**
> > >
> > > Your understanding is correct. We'll follow your suggestions to improve the presentation in both Figure 2 and text.
> > >
> > >
> > > **Design considerations**
> > >
> > > Correct.
> > >
> > >
> > > **Random seeds and statistical significance**
> > >
> > > Indeed, many other papers didn't report the standard error. We'll compare statistical significance as much as possible.
> > >
> > >
> > > **Extent to which results can be explained by increased depth**
> > >
> > > We are glad that your broader concern is unlikely :)
> > >
> > >
> > > **Relationship to indirection in prefrontal cortex / basal ganglia**
> > >
> > > Thank *you* for referring to this paper :)

---

> > > > ### Comment · Reviewer_yiWz · 2021-08-12
> > > > **Score updated**
> > > >
> > > > Thanks to the authors for further clarifications. They have addressed all of my concerns. I have also read through the other reviews and replies. Given that the discussed revisions are implemented, I believe the paper is ready for publication. I am updating my score to a 7.

---

> > > > > ### Author Response · Authors · 2021-08-12
> > > > > **Thanks for increasing the score!**
> > > > >
> > > > > Thank you for increasing the score after reading the other reviews and our replies!

---

### Official Review · Reviewer_VoSb · 2021-07-17

**Rating:** 5
**Confidence:** 4

**Summary:**

The paper proposes a new type of recurrent unit, analogous to GRU and LSTM, dubbed Self-IRU. The underlying update is similar to that used by LSTMs and GRUs. However, the authors suggest to additionally incorporate the output of a different Self-IRU in the forget and output gates, similar to stacking LSTMs or GRUs in multiple layers. The authors propose that this allows the model to learn a "dynamic soft recursion". The paper performs empirical experiments on pixel-wise image classification, a synthetic logical inference task, sorting, tree traversal, music modeling, and semantic parsing tasks.

**Limitations And Societal Impact:**

The paper checklist cites Section 3.7 for discussion of its limitations. Section 3.7 discusses how the soft depth gate values vary rhythmically for a music dataset and "according to the more complex color-image information" for CIFAR-10. Especially for the music dataset, it would be useful if the paper can demonstrate that the varied gate values somehow corresponds to discernible intervals within the music, or if a similar conclusion can be reached on some kind of synthetic dataset.

**Main Review:**

## Originality
The recurrent unit is similar to existing units like GRU and LSTM, but with novel components. In particular, the idea of using previous layers as inputs into the forget and output gates appears to be novel. However, it doesn't seem exceptionally different from the common practice of stacking recurrent networks, although one important difference is that Self-IRU only uses the output of the previous layer for the forget and output gates, rather than throughout the unit.

## Quality
- The paper has very extensive empirical results comparing the proposed recurrent unit to previous work, showing that it can work well in practice.
- It seemed odd that the previous layer's output is used directly for the forget and output gates, rather than having separate linear projections for the gates. More generally, it would improve the paper if there was more explanation about how it was decided where to incorporate the output of the previous layer inside the computation of the current layer.
- I believe that the framing of the work as recursion is not appropriate. Typically, recursion in computer science involves taking a larger input, reducing or dividing it into some way for the recursive call, and then combining the outputs of the smaller problems. There are no such elements of recursion in this work. The recurrent unit proposed in this work does not follow this kind of structure, and instead more closely resembles the use of depth in neural networks.
- There should be a more direct comparison of how the work conceptually differs from stacked recurrent networks. If you have a multi-layer LSTM, then each layer would also use the previous layer's output in computing the forget and output gates, as in this work.
- The ablation studies in Section 3.6 suggest that the relationship between max depth, base transformation, and the resulting performance seems scattered and unintuitive. It would be preferable if such experiments could be crafted to convincingly show that the additions in this work compared to previous work result in benefits, or if it were possible to explain why  Currently, it seems like a portion of the empirical improvement exhibited by Self-IRU could be attributed to more degrees of freedom in hyperparameters allowing for a broader hyperparameter search.

## Clarity
The paper is clear, although I would prefer if some of the framing was different, as mentioned above.

## Significance
Self-IRU appears fairly straightforward to implement so I think others can easily build upon the work, both for applications and for future designs on recurrent units. It's not clear if the empirical improvement is large enough over existing alternatives so that it would see significant use in applications.

**Time Spent Reviewing:**

4

---

> ### Author Response · Authors · 2021-08-10
> **Response to Reviewer VoSb**
>
> Thank you for the insightful comments.
>
> ## On using the previous output for the forget and output gates (bullet point 2)
>
> This is a very smart question. In the LSTM-based architecture in (2.1)--(2.5), linear projections possibly exist in (i) the forget gate $\mathbf{f}_t^{(l)}$; (ii) the output gate $\mathbf{o}_t^{(l)}$; (iii) the candidate memory cell $\mathbf{z}_t^{(l)}$. Our early proof-of-concept experiments evaluated different choices of these three components for introducing the dynamic soft recursion design, showing that the best results are always obtained via choosing both (i) and (ii). We think this is reasonable: the challenge of long-term information preservation and short-term input skipping in latent variable models is addressed by LSTMs, via the control using both the forget gate and the output gate; thus, increasing architectural flexibility in these two critical components (see analysis in Section 3.7) of LSTMs may lead to higher effectiveness, which is demonstrated in the "very extensive empirical results" (thank you for acknowledging this in your bullet point 1).
>
>
> ## On "recursion" (bullet point 3)
>
> According to the definition of "recursion" in the Merriam-Webster dictionary (https://www.merriam-webster.com/dictionary/recursion):
>
> ```
> 1: RETURN
>
> 2: the determination of a succession of elements (such as numbers or functions) by operation on one or more preceding elements according to a rule or formula involving a finite number of steps
>
> 3: a computer programming technique involving the use of a procedure, subroutine, function, or algorithm that calls itself one or more times until a specified condition is met at which time the rest of each repetition is processed from the last one called to the first
> ```
>
> The above definitions (both 2 and 3) fit the idea of "recursion" in our method. For reducing or dividing the original problem via the recursive call, we may also take lower-level recursion nodes as solving smaller problems while the highest node (closest to the input) solves the original problem in the recursive tree (Figure 2). Nonetheless, we are open to use other words (e.g., dynamic soft recursion -> dynamic soft invocation, or any better naming you suggest) and this does not affect our contributions :)
>
>
> ## On comparison with multi-layer LSTMs (bullet point 4)
>
> Conceptually, in sharp contrast to multi-layer LSTMs, our method enables a tree structure of self-instantiation (Figure 2), where the extent of recursion is dynamic (Section 3.7). Empirically, Table 3 compares our method with the stacked BiLSTM. To obtain the best results for both architectures, our method uses a max depth of 2 and the stacked BiLSTM uses 3 (if used 2, results are worse) stacked layers. Overall, our method is the best performer in both the sorting and tree traversal tasks. We will highlight these differences in the revision.
>
>
>
> ## On the empirical improvement exhibited by Self-IRU (bullet point 5)
>
> Section 3.6 shows that the optimal choice is task dependent. This empirically supports our design that the Self-IRU is agnostic to the choice of base transformations (Section 2.3). Besides, different from previous work, in the Self-IRU the extent of recursion is dynamically learned across the temporal dimension of sequences. This inductive bias is analyzed in two aspects in Section 3.7. Overall, these two sections demonstrate the architectural flexibility of the Self-IRU in practice, which may explain the empirical improvement.
>
>
> ## On demonstrating that the varied gate values correspond to discernible intervals within the music (limitations)
>
> Thank you for the suggestions. The music data is in the processed format (see .mat files at our anonymous link: https://github.com/anonymous-sc/Self-IRU/tree/main/SelfIRU/poly_music/mdata), making it hard to tell discernible intervals within the music.

---

> > ### Author Response · Authors · 2021-08-19
> > **Response to Reviewer VoSb**
> >
> > Please do let us know if you have any further questions after reading our response. Thanks.

---

> > ### Comment · Reviewer_VoSb · 2021-08-24
> > **Reply to response**
> >
> > Thank you for your well-reasoned response.
> >
> > *Forget and output gates*: in your response, you mentioned "showing that the best results are always obtained via choosing both (i) and (ii)". Does this mean that there are some kind of linear projections after all in (2.1) and (2.2)? I think that conflicts with what is in the paper, since the paper has
> > $$f_t^{(l)} = \sigma(\alpha_t^{(n)} \odot \text{Self-IRU}^{(l - 1)}(x_t, h_{t-1}^{(l-1)}) + \cdots$$
> > (lacking a further transformation on the Self-IRU output)
> > rather than something like
> > $$f_t^{(l)} = \sigma(\alpha_t^{(n)} \odot (W \cdot \text{Self-IRU}^{(l - 1)}(x_t, h_{t-1}^{(l-1)}) + b) + \cdots$$
> > where the Self-IRU output is transformed.
> >
> > Anyway, what is most important is that the paper is clear. There is nothing particularly wrong with the version currently in the paper, but obviously any insight into the choices made in the paper would be welcome, such as a further description of the early proof-of-concept experiments.
> >
> > *Recursion/comparison with multi-layer LSTMs*: I withdraw my objection about "recursion" also considering that no other reviewers commented about this.
> >
> > If you draw a diagram for the network where you have the input $x_t$ at the bottom, $L$ blocks for the Self-IRUs, then the final output at the top, like
> > ```
> > h_t
> > +------------------+
> > | Self-IRU layer 3 |
> > +------------------+
> > +------------------+
> > | Self-IRU layer 2 |
> > +------------------+
> > +------------------+
> > | Self-IRU layer 1 |
> > +------------------+
> > x_t
> > ```
> > but with the arrows added, I think it is clearer that the proposed architecture is similar to a stacked LSTM but with many skip connections (in particular, x_t is provided to every layer directly), and I believe such a presentation can help readers understand the work in context.
> >
> > *Empirical improvement*:
> > I think "the optimal choice is task dependent" and "the Self-IRU is agnostic to the choice of base transformations" are contradictory. Indeed, the optimal choice being task-dependent shows that the results you obtain with Self-IRU is quite sensitive to which base transformation is used.  I do believe that the work would be stronger if less hyperparameter tuning was necessary.

---

> > > ### Author Response · Authors · 2021-08-25
> > > **Further Response to Reviewer VoSb**
> > >
> > > Thank you for your detailed comments. Below are our further point-by-point clarifications:
> > >
> > >
> > > > "Does this mean that there are some kind of linear projections after all in (2.1) and (2.2)?"
> > >
> > > Per our earlier response, "In the LSTM-based architecture in (2.1)--(2.5), linear projections possibly exist in (i) the forget gate $\mathbf{f}_t^{(l)}$; (ii) the output gate $\mathbf{o}_t^{(l)}$; (iii) the candidate memory cell $\mathbf{z}_t^{(l)}$." To be more concrete, such linear projections possibly exist in $F_f(\mathbf{x}, \mathbf{h})$ in (2.1) and $F_o(\mathbf{x}, \mathbf{h})$ in (2.2), rather than on the Self-IRU output.
> > >
> > > Thanks for pointing out that "there is nothing particularly wrong with the version currently in the paper". Nonetheless, we noted that it's common in gated RNNs to perform the element-wise multiplication between a gate (e.g., our $\boldsymbol{\alpha}$ and $\boldsymbol{\beta}$) and a variable that lacks a further linear transformation (e.g., our Self-IRU output)---see (9.1.2) in https://d2l.ai/chapter_recurrent-modern/gru.html and (9.2.3) in https://d2l.ai/chapter_recurrent-modern/lstm.html. While a further linear transformation on the Self-IRU output also increases parameterizations and computations, we would like to leave a full exploration of this direction to future work.
> > >
> > >
> > > > "I withdraw my objection about 'recursion'"
> > >
> > > Thanks for your open-mindedness!
> > >
> > >
> > > > "Such a presentation can help readers understand the work in context"
> > >
> > > Thanks and we will improve our presentation to more clearly show how Self-IRUs are related to and distinct from multi-layer LSTMs in the revision.
> > >
> > >
> > >
> > > > "'The optimal choice is task dependent' and 'the Self-IRU is agnostic to the choice of base transformations' are contradictory"
> > >
> > > We respectfully disagree. In general, "agnosticism" means "not sure" rather than a certain "yes" or "no". Saying that "the Self-IRU is agnostic to the choice of base transformations" means that no base transformation is a certain "yes" or "no" for the choice, and implies that the optimal choice is task dependent. Moreover, this design allows the Self-IRU to serve as a broader framework that subsumes some other recurrent models. For example, when the base transformation chooses a linear projection, partially linear projection, convolution, and RNN unit, Self-IRUs may generalize standard gated RNNs, SRUs [Lei et al., 2018], Quasi RNNs [Bradbury et al., 2016], and RCRNs [Tay et al., 2018], respectively (See Section A.3 in our supplementary material). Furthermore, note that only two base transformation choices are considered in this work so the hyperparameter tuning is totally feasible.
> > >
> > >
> > >
> > > **Summary**
> > >
> > > Besides all your constructive suggestions, thank you for acknowledging the "novel components", "nothing particularly wrong", and "very extensive empirical results".
> > >
> > > We hope that you will re-assess the merits of our paper with such perspectives in mind. Please do let us know if you still have any major concerns that lead to an unfavorable consideration. We are more than happy to address them further. Thanks!

---

### Official Review · Reviewer_nEcJ · 2021-07-19

**Rating:** 7
**Confidence:** 3

**Summary:**

This paper proposes a new architecture for sequence modelling. The idea is roughly: rather than have stacked layers as in a stacked LSTM, to have an adaptive recursion at each time step. The paper states the equations for this model then mainly presents the performance on a range of problems.

**Limitations And Societal Impact:**

Yes

**Main Review:**

It's hard to evaluate this sort of paper. The writing is very clear and the presentation is good. But on the other hand, it is relatively easy to present what is mainly a set of fairly straight-forward experimental evaluations.

But, this is not to detract from the creativity of the model. The idea is compelling, and the performance looks interesting. But then again, no code is provided, and the results are therefore not easily reproducible and must be taken with a grain of salt. As the paper is purely experimental this is a key consideration. The response to checklist question 3A ("Did you include the code, data, and instructions needed to reproduce the main experimental results") in the paper checklist is yes, and yet ... there is no code. I would increase my review score if the authors could provide, say, an anonymous download link to code that reproduces the main SOTA results.

It is also unclear if the experimental comparisons are appropriate. One obvious example is "Nested LSTMs" by Moniz and Krueger. This paper has a similar idea (albeit with what looks to me like a less compelling execution than that of the paper under review) but is not compared with and not even cited.

**Time Spent Reviewing:**

2

---

> ### Author Response · Authors · 2021-08-10
> **Response to Reviewer nEcJ**
>
> Thank you for the positive assessment, insightful comments, and being specific in questions for increasing your score.
>
>
> ## On source code
>
> Please find our source code at the anonymous link: https://github.com/anonymous-sc/Self-IRU. We would like to leave polishing code/doc and adding more experiments as TODO items.
>
>
> ## On the paper "Nested LSTMs"
>
> Thank you for providing this example. The selected 20+ baselines for comparison are the most representative ones in our investigated tasks. Even among the "Nested LSTMs" paper and its citations, none evaluated nested LSTMs in our investigated tasks (let alone massive other papers without citing it). Among many differences between Self-IRUs and nested LSTMs, a key contribution of our method is that the extent of recursion is dynamically learned across the temporal dimension of sequences, which is impossible by nested LSTMs. We will cite the "Nested LSTMs" paper and compare their differences in the revision.

---

### Official Review · Reviewer_kd1W · 2021-07-20

**Rating:** 6
**Confidence:** 3

**Summary:**

The authors introduce a recursive neural network architecture, using a novel kind of unit termed a 'self-instantiated recurrent unit'.  This architecture enables the gate parameters within an LSTM-like architecture to themselves be determined by units at a lower recursive layer, and the level of recursion to vary dynamically.  The authors test the model empirically on multiple tasks, including image classification, logical inference and music sequence modeling.

**Ethical Concerns:**

Not relevant.

**Limitations And Societal Impact:**

Not relevant.

**Main Review:**

The model architecture is an interesting novel variation on the LSTM architecture.  The ability to dynamically learn the recursive depth is also an interesting idea, and as the authors propose, may be interpretatively relevant In various domains.  The presentation is clear, and the authors offer a thorough experimental investigation of the architecture on multiple domains, achieving state-of-the-art results on several datasets, along with an ablation study, and analysis of the variation in recursive depth across domains, and comparisons with alternative architectures.

Minor points:

1. A potentially interesting comparison would be to compare the relative strengths of including more layers of recursive units versus more layers of 'stacked' units, i.e. whose h's do not connect back to the gate parameters, but feedforward to an output layer.  It would be possible to combine recursive and stacked layers in the same architecture, potentially with the recursive depth varying with the stack layer.

2. The manuscript should be checked for typos before the final version.

**Time Spent Reviewing:**

2 hours

---

> ### Author Response · Authors · 2021-08-10
> **Response to Reviewer kd1W**
>
> Thank you for the positive assessment and insightful comments.
>
> ## On minor points of comparing and combining the relative strengths
>
> We have compared our method (recursive units) with the stacked BiLSTM (stacked units) in Table 3. To obtain the best results for both architectures, our method uses a max depth of 2 and the stacked BiLSTM uses 3 (if used 2, results are worse) stacked layers. Overall, our method is the best performer for both the sorting and tree traversal tasks in Table 3, demonstrating the relative strength of recursive units over stacked units.
>
> It is a very smart idea to combine both architectures. However, our main goal in this paper is to propose a way of recursive self-instantiation via RNN gating functions to enable dynamic soft recursion depth. Since combining both arthitectures may require heavier computations, we would like to leave a full exploration of this direction to future work.
>
>
>
> ## On minor points of checking for typos
>
> Thank you for the suggestions. We will carefully check for typos before the final version. Please let us know if you spot any typos and we will fix them.

---

> > ### Author Response · Authors · 2021-09-01
> > **Response to Reviewer kd1W**
> >
> > Please let us know if our response on Aug 10 addresses your questions or there is anything you'd like to discuss further. Thanks.

---

### Decision · Program_Chairs · 2021-09-27

**Decision:**

Accept (Poster)

**Comment:**

This paper introduces a self-instantiated recurrent unit that is related to the LSTM but with additional capabilities for soft recursive. The authors evaluate their method on a range of tasks including image classification, logical inference, sorting, tree traversal, music modeling, semantic parsing, and code generation.

The reviewers and I agree that the evaluations are quite extensive. It's also clear the method performs well. The model presentation is mostly clear, but there were still a number of queries and points of confusion that popped up in the back-and-forth (relation to stacked LSTM, why only hidden and output gates determined recursively, which parameters are shared, etc.). The author rebuttal helped in this regard, but these points weren't completely resolved. I agree with R-VoSb that the ablation analysis isn't as revealing as it should have been in illuminating the architecture choices. Also, I found that section 3.7 wasn't very developed and didn't add much understanding. I wish it was clearer why and how the architecture works.

That said, I recommend acceptance, based on the strength of the experiments and the cleverness of the architecture. I hope the exposition will be further improved in the final version, as there are many helpful comments from the reviewers.